# Gene Regulation Mediated by microRNA-Triggered Secondary Small RNAs in Plants

**DOI:** 10.3390/plants8050112

**Published:** 2019-04-26

**Authors:** Felipe Fenselau de Felippes

**Affiliations:** Science and Engineering Faculty, Queensland University of Technology, Brisbane, Australia; felipe.felippes@qut.edu.au

**Keywords:** tasiRNA, phasiRNA, miRNA, secondary siRNA

## Abstract

In plants, proper development and response to abiotic and biotic stimuli requires an orchestrated regulation of gene expression. Small RNAs (sRNAs) are key molecules involved in this process, leading to downregulation of their target genes. Two main classes of sRNAs exist, the small interfering RNAs (siRNAs) and microRNAs (miRNAs). The role of the latter class in plant development and physiology is well known, with many examples of how miRNAs directly impact the expression of genes in cells where they are produced, with dramatic consequences to the life of the plant. However, there is an aspect of miRNA biology that is still poorly understood. In some cases, miRNA targeting can lead to the production of secondary siRNAs from its target. These siRNAs, which display a characteristic phased production pattern, can act in *cis*, reinforcing the initial silencing signal set by the triggering miRNA, or in *trans*, affecting genes that are unrelated to the initial target. In this review, the mechanisms and implications of this process in the gene regulation mediated by miRNAs will be discussed. This work will also explore techniques for gene silencing in plants that are based on this unique pathway.

## 1. Introduction

MicroRNAs (miRNAs) are molecules that play pivotal roles in the control of gene expression, and together with small-interfering RNAs (siRNAs) they form the two major classes of regulatory small RNAs (sRNAs) in plants. Biogenesis of miRNAs relies on the activity of DICER-LIKE 1 (DCL1), an RNAse III enzyme that processes transcripts with imperfectly, self-complementary foldback structures to 21–22 nt long mature miRNA. The miRNA is then loaded into ARGONAUTE (AGO), conferring sequence specificity to the RNA-induced silencing complex (RISC), which promotes cleavage of the target transcript through the slicing activity of AGO. Alternatively, RISC-mediated gene downregulation can also be achieved via translation inhibition, a process still poorly understood in plants [1].

There are many examples of physiological and developmental pathways regulated by the direct action of miRNAs [2], yet there is an aspect of miRNA activity that only now has become more evident, and that is the ability of these molecules to indirectly regulate gene expression through the production of secondary siRNAs. In most cases, the outcome of miRNA-loaded RISC activity is cleavage and subsequent degradation of the target transcript, however, in a few cases, targeting can result in synthesis of a double-stranded RNA (dsRNA), having the target transcript as template for RNA-DEPENDENT RNA POLYMERASE 6 (RDR6) [3,4,5,6,7]. This step also requires the activity of SUPPRESSOR OF GENE SILENCING 3 (SGS3) and SILENCING DEFECTIVE 5 (SDE5) [3,4,5,6,7,8,9]. The newly synthetized dsRNA molecule is primarily the substrate for DCL4 to generate a new population of “secondary siRNAs”, which have a phased pattern as their main characteristic, with siRNAs being produced in intervals 21 or 24 nt from the miRNA cleavage site [4,5,7,10,11,12]. These secondary siRNAs can have a dramatic impact on gene regulation mediated by miRNAs. They can act in *cis*, amplifying the silencing effect on their targets, or in *trans*, promoting downregulation of genes that otherwise would not be targeted by the trigger miRNA. Moreover, if the secondary siRNA precursor is a member of a gene family or shares sequences with other transcripts, silencing could spread to other genes, creating a regulatory cascade initially triggered by a single miRNA targeting event [13,14]. In addition to these features, generation of secondary siRNAs by miRNAs can add several other advantages to the regulation of gene expression, which will be discussed in more detail in this review.

The first miRNA-triggered secondary siRNA-producing loci were initially identified and characterized in *Arabidopsis thaliana*, where non-coding RNA molecules were found to give rise to siRNAs suppressing the expression of genes that were unrelated to their precursor molecules; therefore, they were referred as *trans*-acting siRNAs (tasiRNAs) [3,4]. To date, four families of tasiRNA-producing loci (*TAS1-4*) have been described in *A*. *thaliana*. *TAS1* and *TAS2* are both targeted by miR173, with *TAS3* and *TAS4* tasiRNA biogenesis being dependent on miR390 and miR828, respectively [3,4,5,6,15]. Additional *TAS* genes (*TAS5-10*) have been described or predicted in species other than *A. thaliana*, suggesting that many secondary siRNA-producing loci are yet to be discovered [16,17,18,19]. Indeed, with the advance of genomic-scale analyses, several phased, secondary siRNA-producing loci were recently identified in various plant species. As for *TAS* transcripts, one or more miRNAs were shown or predicted to target the precursor RNA molecule. However, different from classic tasiRNAs, generation of these newly identified secondary siRNAs can also be associated with protein-coding genes, and their activity in promoting cleavage of their target in *trans* is often not shown. Therefore, these secondary siRNAs are called phased siRNAs (phasiRNAs), and the loci where they come from is referred to as a *PHAS* gene [20,21]. In summary, phased, miRNA-triggered secondary siRNAs are generally referred to as phasiRNAs, while tasiRNAs are a specialized subclass of phasiRNAs for which function has been demonstrated to occur in *trans*. In addition, *TAS* loci are usually considered as noncoding with no function other than being precursor molecules to secondary siRNAs [13,20,21]. This review will focus on the factors leading to miRNA-triggered production of secondary siRNAs as well as the main features and possible advantages of this system for control of gene expression in plants. In addition, different methods to trigger gene downregulation using this silencing mechanism will be discussed.

## 2. Biogenesis of miRNA-Triggered Secondary siRNAs

The question of how some miRNAs can trigger the production of secondary siRNAs from their targets has been one of the major subjects of phasiRNA research. Two hypotheses have been commonly used to explain this peculiar phenomenon, known simply as “one-hit” and “two-hit” models. However, recent findings have reshaped our understanding of how miRNA-triggered siRNAs are generated (Figure 1). It is worth noting that, despite most of what is known has originated from studies using *TAS* loci as a model, the mechanisms for miRNA-triggered secondary siRNA biogenesis seems to be valid for the majority, if not all, phasiRNAs.

### 2.1. “Two-Hit” Model

The first attempt to explain biogenesis of phasiRNAs came from the observation that *TAS3* in *Physcomitrella patens* (*PpTAS3*), unlike most of the plant miRNA targets, displayed two miR390 complementary sequences, and the majority of sRNAs produced from this transcript were confined between these two sites [22]. The authors also showed that this pattern in the *PpTAS3* gene could be extended to *A. thaliana* and several other species. Interestingly, the miR390 target site located 5’ to the tasiRNAs was not cleavable in *A. thaliana*; however, mutations disrupting this or the cleavable 3’ miR390 site resulted in plants showing phenotypes associated with the impairment of *TAS3* function [22]. This and the discovery that other secondary siRNA-producing loci are also flanked by sRNA complementary sites led to a model where dual sRNA hits would act as a trigger for recruitment of RDR6.

### 2.2. “One-Hit” Model

The “two-hit” model, however, was not sufficient to explain biogenesis of secondary siRNAs from other loci, such as *TAS1*, *TAS2* and *TAS4*, in which tasiRNA precursors were all cleaved at a single site upstream of the sRNA production region [3,4,5,6,15,22]. One of the initial insights into the mechanism behind tasiRNA generation from these transcripts came from experiments testing the requirements for secondary siRNA biogenesis from *TAS1* in *A. thaliana* [23,24]. These authors have shown that miR173-mediated targeting was not only necessary, but also sufficient to trigger tasiRNA production. This observation suggested that miRNAs, such as miR173, had unique features that differentiated them from the majority of other miRNAs that could not trigger the production of secondary siRNAs from their targets. Indeed, it was later demonstrated that miRNA length and the structure of the miRNA/miRNA* duplex (miRNA* refers to the sequence complementary to the predominant miRNA in the precursor molecule) were key determinants in triggering miRNA-dependent secondary siRNA production [25,26,27]. In plants, most miRNAs are processed as 21 nt long molecules. Interestingly, genome-wide analysis of sRNAs found that the majority of miRNAs and siRNAs associated with secondary siRNA production were 22 nt in length, indicating that sRNA size could be a crucial aspect for phasiRNA biogenesis. To test this hypothesis, Cuperus et al. [26] and Chen et al. [25] engineered miRNA precursors to produce mature miRNAs, either 21 or 22 nt of length, and tested their ability to initiate secondary siRNA production. For instance, miR173, which initiates tasiRNA production in *TAS1* and *TAS2*, is naturally found as a molecule 22 nt in length; however, its ability to trigger tasiRNA biogenesis was abolished when this molecule was 21 nt in length. Accordingly, turning the 21 nt miR319, which does not trigger secondary siRNAs production, into a 22 nt miRNA conferred this molecule the capacity to start siRNA generation from its target transcript. In the vast majority of cases, the generation of 22 nt miRNAs, instead of the more commonly found 21 nt variety, is caused by the presence of an asymmetric bulge in the pairing between miRNA and miRNA* in the precursor molecule, resulting in maturation, by DCL1, of an miRNA/miRNA* duplex with a 22/21 nt configuration [25,26]. Interestingly, 22 nt miRNAs can also be created by post-processing modification events, as shown in the case of the soybean miR1510. This sRNA, which is able to trigger phasiRNA production, is processed as 21 nt molecule, but accumulates as a 22 nt isoform via monouridylation [28]. Despite this, it is not only miRNA size that seems to account for the ability to initiate transitivity (another name for RDR6-dependent secondary siRNA production). MiRNAs that are 21 nt in length can also trigger transitivity when their miRNA* is found as a 22 nt molecule. It has been proposed that asymmetry in the miRNA/miRNA* duplex, which is also found in miRNAs processed as 22 nt molecules, is sufficient for the initiation of secondary siRNA production in target transcripts. This idea was confirmed by producing an asymmetric miR173/miR173* duplex, where both miRNA and miRNA* were produced as 21 nt entities, demonstrating that this configuration could efficiently trigger transitivity [27].

### 2.3. A Unified “One-Hit” Model

The models described above have been considered as two independent mechanisms leading to phasiRNA generation in plants. However, a recent study has shown that this might not be the case, and these two processes might be more similar than previously suspected. In *A. thaliana*, miR390 was recently shown to trigger tasiRNA production from the *TAS3* transcript even when only one targeting event occurred, similar to what happened with other *A. thaliana TAS* families [29]. Supporting the idea that “one-hit” is sufficient for tasiRNA production in *TAS3*, many dicots, conifers, and cycads carry a second *TAS3*-related gene, referred to as *TAS3-2*, which, in some species of citrus, chicories, and populous, possess only one miR390 target site [30,31]. In addition, *TAS3* in spruce has been characterized as a large family with 18 members, some of them carrying only one miR390 complementary sequence [32]. Taken together, these observations suggest that “one-hit” might be the basic system behind secondary siRNA production, and the “two-hit” configuration may have evolved as a regulatory mechanism to avoid possible off-targeting incidents by limiting the region from where secondary siRNAs are produced [29]. Another peculiarity of this unified model concerns cleavage of the precursor transcript as a requirement for secondary siRNA generation. Since the discovery of tasiRNAs, the slicing activity of AGO within RISC has been considered essential for the recruitment of RDR6 [13,14,33]. However, secondary siRNAs deriving from *TAS1* and *TAS3* can also be detected, even when the respective transcripts are not cleaved by AGO, indicating that a non-cleavable interaction of RISC with its target is sufficient to trigger efficient phasiRNA production [29,34]. Nonetheless, slicing of the target transcript is still crucial for the proper phase and, therefore, function of tasiRNAs, which is probably due to the lack of a well-defined end of the dsRNA molecule caused by the absence of cleavage.

Despite unification under a same “one-hit” process, tasiRNA biogenesis from *TAS3* and *TAS1/2/4* still differs regarding the initiation mechanism. While *TAS1*, *TAS2*, and *TAS4* give rise to secondary siRNAs after being hit by miRNAs that are 22 nt in length, miR390, which targets *TAS3*, is 21 nt long and does not show asymmetric structures in its precursor [3,4,5,6,15,25,26]. Another particularity of the *TAS3*/miR390 system is that miR390 is “assigned” with its own AGO protein. In *A. thaliana* there are 10 AGOs (AGO1-10), with most miRNAs loaded in AGO1, including the bulk of those that can trigger transitivity [33,35,36]. However, miR390 is not only preferentially found associated with AGO7, but it is also the specific ligand of this protein, which seems to select miRNA through recognition of its initial 5’ adenosine residue and the central region of the miR390/miR390* duplex [33,37]. More interestingly, unlike when it is loaded into AGO7, the ability of miR390 to initiate tasiRNA production was abolished when this miRNA was found or forced to interact with AGO1 or AGO2 [33], highlighting the high degree of specialization found among members of this family.

In summary, in this unified model the production of secondary siRNA from miRNA targets is dependent on a targeting event, where the AGO involved is found in a competent status. This condition is achieved when AGO1, for instance, interacts with an miRNA that is 22 nt long and/or asymmetric as a duplex. AGO7, on the other hand, would be a specialized form of this protein that could continually promote the production of secondary siRNAs from its targets, allowing miRNAs neither 22 nt long nor asymmetric to initiate transitivity. How AGO1, loaded with 22 nt /asymmetric miRNAs or the miR390/AGO7 complex, can route its target to the RDR6 pathway is still unknown. One could speculate that once loaded with 22 nt /asymmetric miRNAs, AGO1 would suffer a change in its configuration allowing the onset of transitivity. Such a change in conformation is supported by crystal structure analysis of *Thermus thermophilus* AGO bound to DNA guide strands of different sizes [38]. In the case of AGO7, it is plausible that this protein has evolved to constitutively be found in this competent form. It is clear that further work will be required to test this hypothesis.

### 2.4. Other Elements Involved in Secondary siRNA Production

The subcellular location where miRNA-triggered secondary siRNAs are produced has also been the focus of investigation. Many components required for phasiRNA biogenesis, such as DCL4, RDR6, SGS3, and AGO7 accumulate in the cytoplasm [39,40,41,42,43]. More specifically, SGS3, RDR6, and AGO7 have been shown to co-localize in cytoplasmic foci, called siRNA bodies, which are distinct from processing-bodies (P-bodies) involved in mRNA turnover [41,42]. In addition, AGO7, miR390, and SGS3 were shown to be present in microsomal fractions and localized in the endoplasmic reticulum (ER), suggesting that phasiRNA production was connected to cytoplasmic membrane structures [40,42]. Indeed, it has been reported that miRNAs, including 22 nt ones, their target transcripts, and AGO1 are found associated to membrane-bound polysomes (MBPs) in the ER. Moreover, miRNA-guided cleavage could also be detected in MBP fractions [44]. Corroborating the view that proper subcellular localization was crucial for miRNA-triggered secondary siRNA production, phasiRNA generation was affected in *ago1-27* plants, most likely from a decrease in association between MBP and AGO1 [44]. Similarly, *TAS3*-tasiRNA biogenesis was impaired when AGO7 was forced to accumulate in the nucleus [42]. It is still unclear how these different components are brought together to the same subcellular compartment, and clearly more research will be necessary.

The role of SGS3 and SDE5 in the production of miRNA-triggered secondary siRNAs is another subject that still remains somewhat enigmatic. SGS3 has been identified from the beginning as an essential component of this system [3,4,5,6,7]. In vitro experiments have shown that SGS3 acts in conjunction with the cleaved transcript, protecting it against degradation and making it available for RDR6 [45]. However, it seems likely that SGS3 has other functions in addition to solely stabilizing the cleaved RNA. As discussed previously, tasiRNA production was shown to be independent of miRNA-mediated cleavage of the precursor transcript, yet SGS3 was still required for the synthesis of secondary siRNAs under non-slicing conditions [29,34]. This scenario is corroborated by the association of SGS3 with a slicing-defective RISC that binds uncut target RNAs [45]. A possible additional function of SGS3 in the production of secondary siRNAs could be in the proper placement of factors involved in transitivity in the same subcellular location, as suggested by the interaction of SGS3 with RDR6 and colocalization with AGO7 in specialized cytoplasmic siRNA bodies [41,42]. As with *sgs3*, the accumulation of secondary siRNAs is also abolished in *sde5* mutants, suggesting a key, although, to date, unclear role for this protein in transitivity [8,9]. SDE5 encodes for a putative RNA export protein, and its role has been suggested to involve the traffic of mRNAs between the nucleus and cytoplasm and/or to route RNA to RDR6 [8,9]. Genetic experiments have placed SDE5 function downstream of SGS3, but upstream of RDR6 activity [46]. Nonetheless, the mode-of-action of these proteins still needs to be investigated in more detail.

In addition to the core components of the pathway, there are other elements that are not essential for the production of miRNA-dependent secondary siRNAs but still have an influence on the biogenesis of these molecules. Components of the THO/TREX complex, which is involved in the intercellular trafficking of mRNAs, have been shown to affect tasiRNA synthesis. In mutant plants where this complex had been disrupted, some tasiRNAs accumulated at lower levels when compared to the wild type [9,47]. It has been suggested that the THO/TREX complex is involved in the transport of *TAS* precursors from their production site to subcellular locations where secondary siRNA biogenesis takes place. Although tasiRNA precursors are considered to be non-coding transcripts, some *TAS* genes have short open reading frames (ORFs) located just upstream of the tasiRNA-producing region, which could potentially give rise to small peptides. Indeed, it has been described that some of these ORFs interact with ribosomes and are actually translated [44,46,48,49]. Interestingly, this process seems to be important for the proper accumulation of tasiRNAs from the transcripts involved. In *TAS2* and *TAS3*, mutations affecting these ORFs result in reduction of tasiRNA accumulation, most likely because of decreased stability of the *TAS* precursor caused by lower levels of association with ribosomes [46,49]. Alternatively, ribosome occupancy has been suggested as a factor that defines the regions of a transcript giving origin to secondary siRNAs [44]. The importance of translation in the production of secondary siRNAs is corroborated by the observation that production of synthetic tasiRNAs (syn-tasiRNAs) is improved with the introduction of a stop codon immediately before the miR173 target site [50].

## 3. Features and Advantages of miRNA-Triggered Secondary siRNA Gene Regulation

In *A. thaliana*, miRNA targeting events leading to transitivity are uncommon, with only a few cases described, and are better exemplified by the *TAS1–4* families. Nonetheless, gene regulation promoted by miRNA-triggered secondary siRNAs have an important impact on plant development and physiology. For instance, miR390 targeting of *TAS3* results in the production of tasiRNAs that can regulate the expression of different auxin response factor genes (*ARFs*), affecting important functions such as leaf morphology, the transition from juvenile to adult phase, and flower and root formation to mention a few [3,5,6,11,51,52,53,54,55]. In recent years, with the popularization and expansion of genomic-based studies, several loci that spawn phased, miRNA-triggered secondary siRNAs were identified in numerous other species. In many cases, these sRNA populations seem to have a role in a variety of pathways related to development, response to stresses, and disease resistance (for more details on these pathways and the miRNAs/phasiRNAs involved, please see this recent review [21]). But what would be the benefits of such an indirect role of miRNAs in the control of gene expression? In the second part of this review, some features and putative added values of indirect gene regulation by miRNAs via secondary siRNAs will be discussed.

The obvious consequence resulting from the production of tasi- and phasiRNAs is the amplification and potential enhancement of the silencing signal (Figure 2A). From a single miRNA targeting event, a population of secondary siRNAs is produced, all with the potential to silence in *cis*, multiplying the number of molecules that could cause downregulation of the precursor loci and, therefore, increase silencing pressure on the target. Indeed, evidence of secondary siRNA targeting in *cis* are quite common. For instance, in watermelon, *Medicago*, and citrus, several phasiRNAs were reported to target their precursor transcripts [20,56,57]. Despite their function mainly being associated with the silencing of unrelated genes in *trans*, tasiRNAs are also known to promote cleavage of the transcript of origin in *cis*. TasiRNA-5D2(-), one of the tasiRNAs emerging from *TAS3*, has been shown to cut its precursor transcript in different species [5,20,22,29,58,59]. In this case however, because it involves a non-coding transcript, it is possible that the re-attack of tasiRNA-5D2(-) acts more as a feedback regulatory mechanism, fine-tuning tasiRNA levels.

Another interesting aspect of gene regulation mediated by secondary siRNAs is the possibility that one miRNA could affect the expression of several genes that otherwise would not have been targeted (Figure 2B). This is well illustrated in cotton, where cleavage of *MYB2* by miR828 results in the production of tasiRNAs that have been predicted to target several unrelated genes, such as sucrose synthase, histone acetyltransferase, and glutamate receptor, none of which are targeted by the triggering miRNA [60]. The *P. patens* miR390/*TAS3* system is an additional case where one miRNA can promote downregulation of several genes that are unrelated, sequence- and function-wise. In addition to the well-characterized, interspecies-conserved, tasiRNA targeting *ARF*-like mRNAs, *P. patens TAS3* transcripts also give origin to secondary siRNAs that can promote downregulation of three AP2 domain-containing transcripts [61]. Alternatively, phasiRNAs can also increase the number of genes regulated by a single miRNA if these secondary siRNAs are produced from conserved regions (Figure 2C). In this scenario, the newly generated siRNAs could function not only in *cis*, but also in *trans*, with the potential to affect any transcript that shares this same conserved region. The best example of such a regulatory network has been described in *Medicago* and involves the generation of phasiRNAs from nucleotide-binding leucine-rich repeat (*NB-LRR*) disease-resistance genes. In this species, three miRNA families (miR1507, miR2109, and miR2118) were described to target different NB-LRR-conserved motifs in 74 transcripts, leading to the biogenesis of phasiRNAs with the potential to regulate 60% of the estimated 540 *NB-LRR* genes [20]. Another mechanism resulting in the expansion of the miRNA activity range is through the production of secondary siRNAs that are 22 nt in size (Figure 2E). In peaches, two miR7122-triggered tasiRNAs, which are predominantly 22 nt in length, have been described to initiate phasiRNA generation from their targets [62], similar to what has been reported for the miR173/*TAS2* pathway in *Arabidopsis* [25].

As discussed previously, mechanisms leading to the formation of phasiRNAs require elements of the miRNA pathway as well as new components, such as RDR6, SGS3, SDE5, and AGO7. This increase in complexity brings new possibilities of regulation with interesting consequences to the indirect function of miRNAs in controlling gene expression. In *Arabidopsis*, tasiRNA production from *TAS3* is dependent on the activity of miR390 and AGO7 [5,33]. Interestingly, the *TAS* transcript, the AGO protein, and miRNA have distinct expression patterns and as a consequence; synthesis of secondary siRNAs from *TAS3* is restricted to cells where all these elements are present (Figure 2D) [33,54,63,64]. This spatio-temporal coordination has been shown to be important for the proper development of leaves. Abaxial/adaxial fate specification is a result of asymmetric expression of *ARFs* in the leaf, caused by the polarized accumulation pattern of *TAS3*-tasiRNAs [63,64,65]. This localized accumulation of tasiRNAs is only possible because of a delimited presence of AGO7 and *TAS3* to the adaxial side, restricting the biogenesis of secondary siRNAs to this region, despite the broader miR390 expression domain [63,64].

With few exceptions, most miRNA precursors are processed by DCL1 into mature molecules 21/22 nt in length, loaded into AGO1, and promote post-transcription gene silencing [1]. However, a whole new level of plasticity can be added to the control of gene expression when miRNA-triggered secondary siRNAs are employed (Figure 2E). The dsRNA molecule synthetized by RDR6 from *TAS* and *PHAS* transcripts are primarily processed by DCL4 into 21-nt-long siRNAs, which like miRNAs, act post-transcriptionally [7,10,11]. Nevertheless, grasses possess an additional DCL enzyme, DCL5 (formerly known as DCL3b), which is responsible for the production of phasiRNAs 24 nt in length from transcripts targeted by miR2275 [66,67]. Interestingly, this is the same size of siRNAs that interact with AGO4, the main effector of the transcriptional gene silence (TGS) pathway, which results in DNA methylation and subsequent silencing [1]. The implications of this discovery are still elusive. These 24 nt phasiRNAs were first described in rice and maize reproductive tissues. They accumulated in meiotic-stage anthers and, therefore, were believed to be involved in reproduction [12,68]. More importantly, given the size of these siRNAs, it is tempting to speculate that 24 nt long, DCL5-dependent phasiRNAs can be associated with AGO4 to promote DNA methylation, adding a new layer to gene regulation mediated by miRNAs. Supporting this view, Xia and colleagues [69] found that the miR2275/24 nt phasiRNAs pathway is not only present in monocots but also in eudicots plants. However, differently from the former group, miR2275-dependent, 24 nt long phasiRNA production in eudicots does not rely on the activity of a specific protein, such as DCL5, but instead it most likely requires the action of DCL3. This is the same enzyme responsible for producing the 24 nt siRNA associated with AGO4 and involved in TGS [1]. Reflecting the high level of specialization and conservation of the pathway, the vast majority of mature miRNAs have a 5’ terminal uridine (U), which has been shown to be a key determinant for the sorting of sRNAs into AGO1 [33,35,36]. In contrast, AGO2 and AGO4 prefer sRNAs that contain an adenosine (A) at the 5’ end, while AGO5 is more often associated to molecules that have a cytosine (C). Many of the secondary siRNAs with conserved functions, such as tasiRNAs that target *ARF* genes, are similar to miRNAs, having an uridine at the 5’ extremity of the mature molecule and, thus, are loaded into AGO1 [5]. Nonetheless, many phasiRNAs do not follow this trend, with many of them found associated to other AGOs such as AGO2 [33,35,36]. In rice, MEL1 is a specialized protein ortholog of AGO5, and it has been described to preferentially bind phasiRNAs that begin with a cytosine [70]. The function of these secondary siRNAs is still poorly understood, but nevertheless, MEL1 has been shown to mediate sporophytic germ-cell development and meiosis, suggesting that these sRNAs might play a direct role in these processes [71]. In summary, the variability of features found among the different AGOs can also be explored when miRNAs that trigger transitivity are involved in gene regulation (Figure 2F).

In addition to the features discussed above, there may be other unknown or poorly understood characteristics of phasiRNAs that could add extra value to miRNA-regulated pathways. For instance, compared to other classes of sRNAs, tasiRNA have been described to display extended cell-to-cell mobility, suggesting that, by initiating transitivity, miRNAs could increase their range of activity (Figure 2D) [72]. Indeed, an artificial miRNA (amiRNA) designed to be produced as a molecule 22 nt in length was reported to start secondary siRNA biogenesis from its target, resulting in silencing in tissues that otherwise would not be affected by amiRNAs of regular size or siRNAs produced from a hairpin construct [73].

## 4. Utilizing miRNA-Triggered Secondary siRNAs to Promote Directed Gene Silencing

Silencing promoted by sRNAs is not only an important mechanism to control gene expression, but it has also been used as a powerful tool to downregulate transcripts in both academic and applied purposes [74]. Despite not being as popular as founding techniques, such as artificial miRNAs (amiRNAs) and hairpin RNA interference (hpRNAi), systems based on the ability of miRNAs to start transitivity also exist and are undoubtedly a valuable addition to the collection of methods aiming to control gene activity (Figure 3).

Artificial tasiRNA (atasiRNA), also known as synthetic tasiRNA (syn-tasiRNA), was the first method developed based on miRNA-triggered biogenesis of secondary siRNAs. It has been used to successfully reduce gene expression of endogenous sequences and to interfere with viroid infections [23,24,33,75,76,77]. This approach consists of replacing one or more tasiRNAs in a *TAS* transcript for sequences devised to target the gene of interest, in a process similar to the design of amiRNAs (Figure 3A) [75,78]. One of the most useful features of this technique is the possibility of having one precursor producing several atasiRNAs, each targeting different sequences, which could be located in the same or distinct transcripts [76]. AtasiRNAs share many of the advantages and limitations of amiRNAs. One distinct advantage is that high levels of specificity can be achieved, decreasing the chance of off-targeting. However, to efficiently design atasiRNA molecules that specifically downregulate one or just a few genes, with minimal chances of silencing unwanted transcripts, it is important that information about the entire genome is made available. Moreover, compared to approaches that make use of whole gene fragments for generation of the silencing construct, such as hpRNAi, this method is more susceptible to the effects of target accessibility that could reduce the effectiveness of the sRNA [79].

An additional system exploring the transitivity initiated by miRNAs is referred to as miRNA-induced gene silencing (MIGS), which has as its main characteristic the easiness of design [80,81]. With a single PCR step, the target site of an miRNA triggering phasiRNA production can be fused upstream of the fragment of a gene of interest. Upon transcript cleavage by RISC, the newly synthetized secondary siRNAs can subsequently promote silencing of related target sequences. MIGS is also a powerful tool to downregulate multiple genes using a single vector (Figure 3B). This is achieved simply by linking fragments of different targets, each with their own miRNA target site [80]. Since this method does not require genome-wide data, it is an interesting alternative to be used in species where this information is still lacking. Despite the aforementioned advantages, the risk of off-targeting needs to be considered when employing this system. Differently to atasiRNAs, MIGS constructs give rise to a population of siRNAs, all with the capacity to silence related sequences. Therefore, depending on the degree of conservation present in the fragment used, genes (other than the intended target) that share similar sequences could also become silenced. To date, MIGS has been shown to be an effective tool to silence genes in several species, including *Arabidopsis*, *Nicotiana benthamiana*, *Medicago*, soybean, rice, and petunia [80,82,83,84,85,86].

A common theme between atasiRNAs and MIGS is the requirement of a trigger miRNA. Therefore, it is important to take into consideration that the spatio-temporal expression pattern of the miRNA could influence the way in which atasiRNAs and MIGS-derived siRNAs are produced. In addition, some species might not code for the miRNA initiating the production of secondary siRNAs from the silencing constructs. To overcome this issue, a collection of plasmids has been created to allow the co-expression of the triggering miRNA and the MIGS/atasiRNA construct from a single vector [80,87]. Alternatively, designing an amiRNA to be produced as a 22 nt long molecule could also be a way to silence genes via secondary siRNAs, without the limitation of a two-component system (Figure 3C). McHale and colleagues [73] have demonstrated that a 22 nt long amiRNA targeting *CHALCONE SYNTHASE* (*CHS*) was able to cause widespread silencing due to the production of secondary siRNAs by RDR6.

## 5. Conclusion and Final Remarks

This review has discussed some of the crucial aspects related to the production of secondary siRNA triggered by miRNA and how this process can add valuable features to the control of gene expression mediated by sRNAs. Moreover, different methods to promote gene silencing in plants that are based on this unique ability of certain miRNAs were discussed, showing that they can be important alternatives to well-established systems, such as amiRNA and hpRNAi.

In recent years, our understanding of the mechanisms leading to miRNA-triggered secondary siRNA generation, and the importance of these molecules in plant physiology and development, has increased rapidly, yet this pathway is still one of the least understood among different processes involving sRNAs. It is still unknown, for example, how 22 nt long miRNA loaded into AGO1 or the miR390/AGO7 complex can lead to the recruitment of RDR6 to the target transcript. Also, the role and molecular mechanisms behind the activity of many phasiRNAs recently described in different plant species remain elusive. The elucidation of these and other aspects related to miRNA-triggered secondary siRNAs will greatly improve our understanding of how sRNAs impact the proper development of plants and the response to abiotic and biotic stresses. In addition, this new knowledge could be useful for the development of new technologies for biotechnological applications.

## Figures and Tables

**Figure 1 plants-08-00112-f001:**
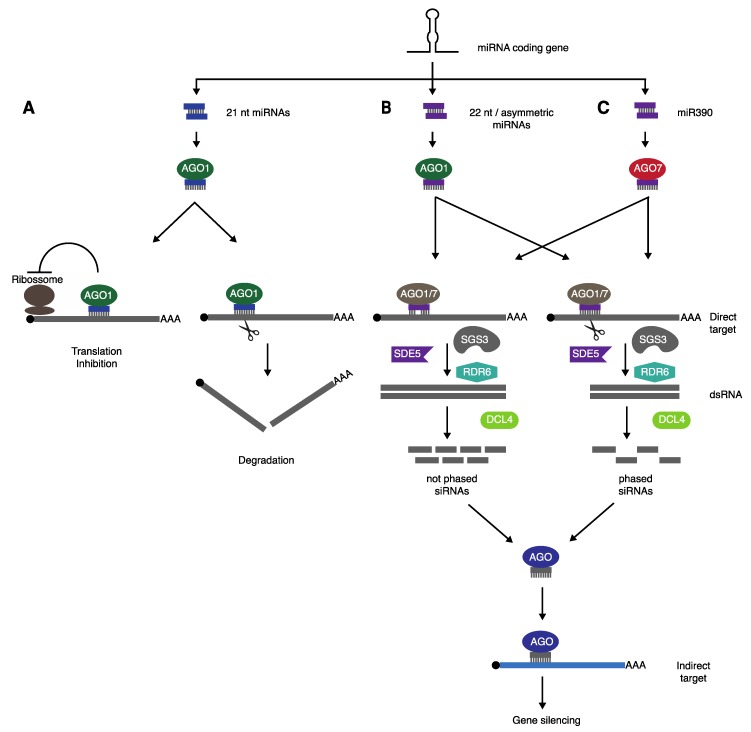
Biogenesis of miRNA-triggered secondary siRNAs. (**A**) Most plant miRNAs are processed into 21 nt long molecules, loaded into AGO1, and promote post-transcriptional gene silencing (PTGS) via translational repression or cleavage, followed by degradation of the target transcript. (**B**) Production of secondary siRNAs occurs when 22 nt long/asymmetric miRNA are bound to AGO1 to target a transcript. (**C**) Alternatively, 21 nt long miRNAs, such as miR390, can also initiate transitivity via interaction with AGO7. In both cases (B and C), cleavage is required for phasing, but not for generation of secondary siRNAs. Biogenesis of secondary siRNAs is dependent on the action of RDR6, SGS3, and SDE5, resulting in the synthesis of a dsRNA, which is mainly processed by DCL4. These siRNAs are loaded into AGOs and can drive gene silencing of their targets.

**Figure 2 plants-08-00112-f002:**
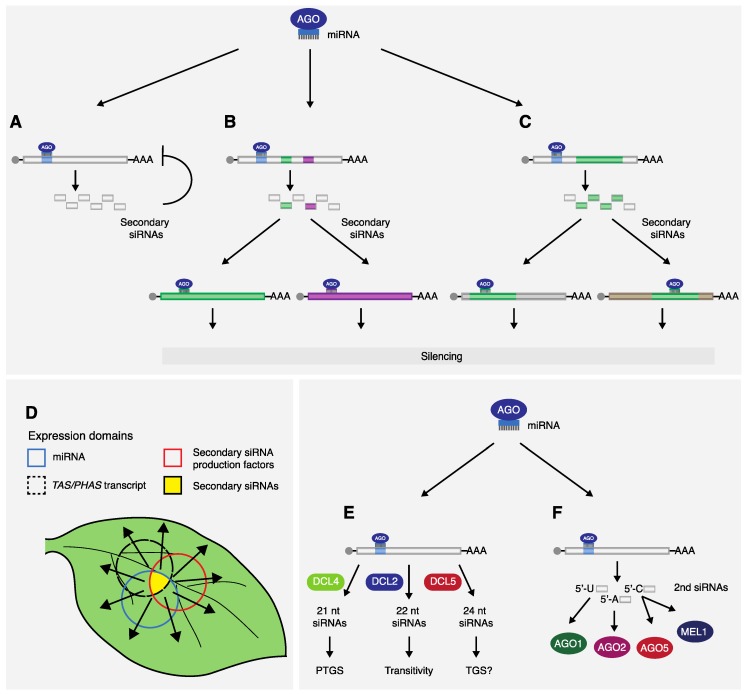
Features and advantages of gene regulation via miRNA-triggered secondary siRNAs. (**A**) By producing secondary siRNAs, miRNAs can increase the silencing pressure on their targets. (**B**) Secondary siRNAs targeting distinct transcripts can be produced from the same precursor, increasing the silencing range of the trigger miRNA. (**C**) The number of genes indirectly regulated by an miRNA can be increased if secondary siRNAs are produced from regions containing a conserved sequence shared by different loci. (**D**) Production of secondary siRNAs is restricted to regions where all elements participating in their biogenesis are present; however, they could later spread to neighboring cells to function non-cell autonomously (as indicated by the arrows). (**E**) The dsRNA synthetized by RDR6 is mainly processed by DCL4, generating 21 nt long siRNAs that are involved in PTGS. Alternatively, the dsRNA can also be the substrate for other DCLs, such as DCL2 and DCL5, resulting in the biogenesis of secondary siRNAs with different characteristics and functions. (**F**) Compared to most miRNAs, which have a 5’ terminal uridine and are loaded into AGO1, secondary siRNAs show an increased diversity on their 5’ extremity, allowing for sorting into different AGO proteins, with possible consequences to their activities.

**Figure 3 plants-08-00112-f003:**
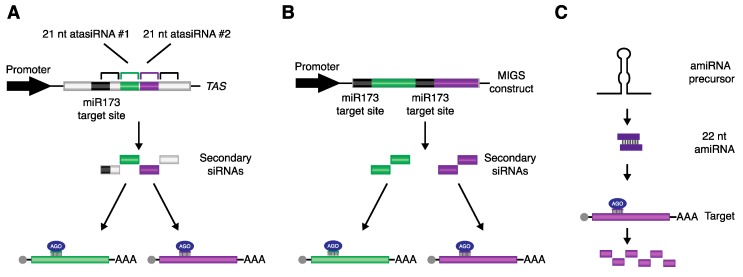
Methods for gene silencing based on miRNA-triggered secondary siRNAs. (**A**) Silencing using atasiRNAs consists of replacing one or more of the tasiRNAs in the *TAS* gene for a sequence designed to target the gene of interest. (**B**) miRNA-induced gene silencing (MIGS) constructs can be generated by placing the sequence recognized by an miRNA that can start transitivity in front of a fragment of the target gene (e.g., miR173). Downregulation of more than one gene using this technique can be easily accomplished by repeating the same pattern with different gene fragments. (**C**) By using specific precursors (such as *MIR173*), 22 nt long amiRNAs can be produced. These molecules can then initiate secondary siRNA synthesis from their targets, adding new features to the original method.

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
