# Peer review of "Gene Regulation Mediated by microRNA-Triggered Secondary Small RNAs in Plants"

_plants, 2019, doi:10.3390/plants8050112_

Round 1
Reviewer 1 Report
The review by Fenselau de Felippes is very well structured and covers extensively the literature. It is overall very well written and the illustrations clear and helpful. The literature is assessed critically and the unknown in the field are nicely identified. This is a very good work.
Yet, it would benefit from a careful proofreading for spelling mistakes and grammar. I noted below a few that caught my attention, there are certainly more:
l190 citoplasmic -> cytoplasmic
l197 essencial -> essential
l201 disrupt -> disrupted
l217 spawning -> spanning
In the section discussing that TAS precursors contain small ORFs (~l190) and the role of ribosomes/translation in the biogenesis of secondary phased siRNA, the papers of Bazin et al. PNAS 2017 and Hou et al. TPC 2016 should be mentioned.
Author Response
Response to Reviewer #1:
I thank the reviewer for the positive response. The reviewer noticed though, a series of grammar and spelling mistakes, which I would like to apologize for. I have corrected these and additional mistakes and, in order to improve the quality of the text, I have had the manuscript checked by a native English speaker colleague. In addition, the reviewer pointed out that two citations were missing from the discussion regarding the impact of small ORFs and translation on the biogenesis of secondary siRNAs. I have, therefore, added these and also the work of Li et al 2016 (Elife) to the manuscript and discussed them appropriately.
Reviewer 2 Report
The review of Fenselau de Felippes summarizes recent findings how miRNAs trigger the production of secondary small RNAs in plants, and how these secondary small RNAs (may) regulate gene expression.
Overall the review is well-structured and well written. There are only minor comments to be addressed. The manuscript contains a number of small mistakes in English spelling and grammar (see below). Furthermore, it is irritating for the reader to find sentences such as
Line 74: This will be discussed in more detail in the next two sections.
Line 221: In this section, some features and putative added values….
Please delete them.
The heading of subchapter 4. should be improved – please better describe what you discuss in the chapter, e.g. manipulation of gene expression based on miRNA triggered secondary siRNAs.
Figure caption 2: It is not clear what the last sentence for Figure legend 2 means (starting line 248 – Compared to ….). The verb “present” does not fit with the text, and it is not clear what is meant – represent? Show?
Line 344-345: please rephrase; the current sentence is not understandable.
Minor but crucial mistakes:
Iine 31 achieved
Line 47 cascade
Line 65 will focus on
Line 140 spruce (no capital letter)
Line 152 well-defined
Line 195 this protein still needs…
Line 221 such an
Line 245 mainly
Line 246 substrate
Line 285 miRNA precursors
Line 342 share
Line 352 synthetized
Line 353 to downregulate multiple
Line 369 to be reproduced
Line 377 based on this
Line 383 into AGO1
Line 384 to the recruitment
Author Response
Response to reviewer #2:
I would like to acknowledge the reviewer for the helpful comments and suggestions. The reviewer has concluded that this manuscript is “well-structured and well written”, with only minor issues that need to be addressed.
Minor comments:
1) “The manuscript contains a number of small mistakes in English spelling and grammar.”
I would like first to apologize for such mistakes. I have corrected these and additional errors. In addition, I have asked a native English speaker colleague to checked this manuscript. I believe that the text quality has now dramatically improved and meets the expected standards of a scientific publication.
2) “It is irritating for the reader to find sentences such as: (Line 74) This will be discussed in more detail in the next two sections or (Line 221) In this section, some features and putative added values” - Please delete them”.
I have deleted or modified the above mentioned sentences. Hopefully, these modifications have now improve the reading experience, without affecting the content or structure of the manuscript.
3) “The heading of subchapter 4. should be improved – please better describe what you discuss in the chapter, e.g. manipulation of gene expression based on miRNA triggered secondary siRNAs”.
The heading of subchapter 4 has been changed. I believe the new heading better describe the content of this section.
4) “Figure caption 2: It is not clear what the last sentence for Figure legend 2 means (starting line 248 – Compared to ….). The verb “present” does not fit with the text, and it is not clear what is meant – represent? Show?”
I have re-written this part of the figure caption.
5) “Line 344-345: please rephrase; the current sentence is not understandable.”
This sentence was modified to make it clearer.
Reviewer 3 Report
This review concisely discussed thebiogenesisof trans-acting siRNAs (tasiRNAs) in plants and the potential application. However, the current manuscript appears to resemble a mini-version of some published reviews, such as the Plant Cell review by Blake Meyers back in 2013 (PMID:23881411). While the topic is certainly interesting, especially with some exciting progress in the field, this manuscript may be more appealing if recent literature will be discussed. I hope the author may consider to include some of the following suggestions for revision.
1) The author used “microRNA-triggered secondary small RNAs in plants”, but excluded most of the phased secondarysiRNAs(phasiRNAs) from the discussion (Page 2, line 66). In this regard, the title does not reflect the content. Also, some of the literature used for discussingtasiRNAbiogenesisareactually forphasiRNAbiogenesisbased on the author’scriterium(Ref 28).
2) Following the first issue, the author seems to considertasiRNAsandphasiRNAsas parallel groups. However, these two groups share the samebiogenesispathway and exhibit similar patterns (both are inphasedpatterns). As discussed by Blake Meyers in his 2013 Plant Cell review,phasiRNAsarelikely a broader group that containstasiRNAs. The only difference is thattasiRNAs(and actually not all but only one or a fewsiRNAsfrom a given locus) have the trans-acting activity. One of Blake’s recent manuscripts clearly states this concept (https://doi.org/10.1101/158832). This concept is gaining more attention and has been stated in multiple recent publications (PMID: 22156213, 25980406, 27900550, 27965387, 28331096, 28852064, 29327403, 29764387, 30458274, 30458274, and so on).
3) Also, one recent publication showed that somephasiRNAscan becometasiRNAs, by acquiring trans-acting activity, upon biotic challenges (PMID: 28331096). This observation further blurs the boundary betweentasiRNAandphasiRNA. In addition, a recent discovery shows that parasitic plants can “inject” theirmicroRNAsto host plants as a trigger to inducephasiRNAproduction (PMID: 29300014). Furthermore, a disease resistance protein SNC1 can suppress thebiogenesisof phasedsiRNAs(PMID:30498229). The point is that phasedsiRNAsare a major component in plant biotic interactions, and the author may consider having some brief discussions.
4) This is a minor point, but it will be helpful to discuss thebiogenesissite of phasedsiRNAsaccording to a new finding (PMID: 27938667).
5) The manuscript is well written, but some small mistakes still exist. For example, Page 2 line 65 “Focus in” should be “focus on”. Page 5 line 181, “SGS3 act” should be “SGS3 acts”.
Author Response
Response to reviewer #3:
Reviewer #3 has contributed with several valuable suggestions, many of them included in the manuscript, which indeed have helped improving the quality of this work (discussed in more detail in the following paragraphs). However, I must disagree with the reviewer’s opinion, that this manuscript resembles a mini version of the excellent Plant Cell review by Blake Meyers group back in 2013 (PMID:23881411). Perhaps, the biggest overlapping between Meyers’ publication and this manuscript is a section regarding the biogenesis of miRNA-triggered secondary siRNAs, which in my opinion should be part of any review about this subject. However, given the six years separating both works and the progress made on the field, the content of this section is quite different. In addition, Meyers’ review only gives an overview of the process, while in the submitted manuscript the different aspects of miRNA-triggered secondary siRNAs biogenesis are extensively covered. The main focus of the publication by Blake Meyers group however, is the discussion about the recent discovery, at the time, of phasiRNAs produced from large families of coding genes (such as PPR, MYB and NB-LRR transcripts). On the other hand, the manuscript I have submitted for publication in Plants focus on three different topics, as follows: a) an extensive review on the biogenesis of miRNA-triggered secondary siRNAs, covering several recent findings posterior to the review of Meyers’ group; b) a systematic discussion about the possible advantages of the gene regulation mediated by phasiRNAs, with a general point of view and based not only in a specific circumstance, such as the case of phasiRNAs originating from large gene families; c) and an analysis of the different methods for silencing genes based on phasiRNAs, including their main characteristics, advantages and limitations.
I also do not share the reviewer’s point of view regarding the significance of this manuscript to the field. In the past few years, several studies have contributed to increase our understanding of how miRNA-triggered secondary siRNAs are generated, a topic which was extensively covered in the submitted work. As mentioned before, this review systematically discuss the advantages and added values of phasiRNAs to the gene regulation mediated by miRNAs, a topic that until now, has been poorly covered. This is of special importance to readers that are not familiarized with the field, helping them to better understand the consequences of this pathway to the control of gene expression. It might also give researchers that are specialists on the topic of sRNA silence, new insights regarding miRNA-triggered secondary siRNAs. Finally, the last part of this manuscript analyze different methods for gene silencing in plants using secondary siRNAs, which, given the importance of gene manipulation in modern biology, could be of interest for many laboratories around the world.
The reviewer also give five the following suggestions/comments for the improvement of the manuscript:
1) The author used “microRNA-triggered secondary small RNAs in plants”, but excluded most of the phased secondary siRNAs (phasiRNAs) from the discussion (Page 2, line 66). In this regard, the title does not reflect the content. Also, some of the literature used for discussing tasiRNA biogenesis are actually for phasiRNA biogenesis based on the author’s criterium(Ref 28).
This comment is also related to suggestion number 2 and it seems to rely on the proper classification of tasi- and phasiRNAs. I am aware and agree with the current classification put forward by Blake Meyers’ group that tasiRNAs are a subclass of specialized phasiRNAs originating from non-coding transcript, which function seems to be mainly to silence genes in trans. I have changed the text accordingly to properly state this fact. Moreover, I refrain to use the term tasiRNA as a synonym for phasiRNAs, unless the example discussed involved specifically one of the 10 known TAS families. In addition, much of what we know about phasiRNAs come from studies using TAS loci as a model, contributing to the impression that parts of this review might focus exclusively on tasiRNAs. In order to resolve this issue, I have added this information to the text. Nonetheless, during the preparation of this review, I have tried to use a variety of examples to illustrate the discussed ideas, especially in the second part, where well-established and newly-characterized examples of phasiRNAs producing loci were used.
2) Following the first issue, the author seems to consider tasiRNAs and phasiRNAs as parallel groups. However, these two groups share the same biogenesis pathway and exhibit similar patterns (both are in phased patterns). As discussed by Blake Meyers in his 2013 Plant Cell review, phasiRNAs are likely a broader group that contains tasiRNAs. The only difference is that tasiRNAs (and actually not all but only one or a few siRNAs from a given locus) have the trans-acting activity. One of Blake’s recent manuscripts clearly states this concept (https://doi.org/10.1101/158832). This concept is gaining more attention and has been stated in multiple recent publications (PMID: 22156213, 25980406, 27900550, 27965387, 28331096, 28852064, 29327403, 29764387, 30458274, 30458274, and so on).
As discussed in the item number 1, I have made changes in the text to properly address this issue.
3) Also, one recent publication showed that some phasiRNAs can become tasiRNAs, by acquiring trans-acting activity, upon biotic challenges (PMID: 28331096). This observation further blurs the boundary between tasiRNA and phasiRNA. In addition, a recent discovery shows that parasitic plants can “inject” their microRNAs to host plants as a trigger to induce phasiRNA production (PMID: 29300014). Furthermore, a disease resistance protein SNC1 can suppress the biogenesis of phased siRNAs (PMID:30498229). The point is that phased siRNAs are a major component in plant biotic interactions, and the author may consider having some brief discussions.
These are indeed very interesting subjects related to the biological role of phasiRNAs in the plant physiology and development, however, as discussed above, this is not the focus of the submitted manuscript. In addition, these and many other examples of the biological role of phasiRNAs in plants have been recently reviewed elsewhere (Deng et al2018, doi: 10.1111/pbi.12882).
4) This is a minor point, but it will be helpful to discuss the biogenesis site of phased siRNAs according to a new finding (PMID: 27938667).
Following the reviewer’s suggestion, I have now included this discussion in the text, also adding the contributions of Glick et al(2008), Elmayan et al(2009), Kumakura et al (2009), Jouannet et al (2012) and Pumplin et al(2016) to this subject.
5) The manuscript is well written, but some small mistakes still exist. For example, Page 2 line 65 “Focus in” should be “focus on”. Page 5 line 181, “SGS3 act” should be “SGS3 acts”.
I apologize for the misspellings and grammar mistakes present in the manuscript. In order to improve the text quality to levels expected in a scientific publication, this work was checked by a native English speaker colleague and newly identified mistakes were corrected.
Reviewer 4 Report
The submitted manuscript reviews the molecular mechanism of biogenesis of secondary siRNAs and their physiological roles in plants. Overall, the review is clearly written and the figures are sufficiently informative. One thing that I thought is that it would be nice if the author could cite recent works (e.g. Xia et al. (2019) Nat Commun, 10, 627) that report the presence of 24-nt phasiRNAs in some of angiosperms and their possible roles in anther and/or pollen development.
Minor comments:
Typos and grammatical errors should be corrected. Some are shown below.
L47, “cascaded” should be “cascade”.
Fig.1 “Ribossome” should be “Ribosome”.
L.190, “citoplasmatic” should be “cytoplasmic”.
L.353, “downregulated” should be “downregulate”.
Author Response
Response to reviewer #4:
I would like to thank the reviewer for the positive reaction and suggestions for the improvement of this manuscript. Despite finding the manuscript well written, the reviewer could identified several grammar and spelling mistakes, for which I apologize. In order to meet the levels of quality expected for a scientific publications, I had the manuscript checked by a native English speaker colleague.
The reviewer has also suggested that this work discuss the presence of 24-nt phasiRNAs in some of angiosperms and their possible roles in anther and/or pollen development, including the research covered in Xia et al. 2019 (Nat Commun, 10, 627). I have accepted this suggestion, discussing not only the formerly mentioned publication, but also the work from Johnson et al 2009 and Zhai et al 2015.
Round 2
Reviewer 3 Report
This revision has taken care of most of my concerns. Overall, it is well organized and readable. I will recommend it for publication. However, I still find numerous grammatical issues. I believe the journal production team will take care of that, so I leave the editor to decide if a minor revision is needed.
Grammatical issues in the first paragraph as an example:
Line 24, "with pivotal role" should be "with a pivotal role" or "with pivotal roles".
Line 26, "aRNAse" should be "an RNase".
Line 27, "that process transcripts" should be "that processes transcripts".
Line 28, "maturemiRNA." shoud be "maturemiRNAs.".
Line 28, "ThemiRNA is then loaded into ARGONAUTE (AGO)" will be better as "ThosemiRNAs are then loaded into ARGONAUTE (AGO) proteins".